# Near real-time adjusted reanalysis forcing data for hydrology

Peter Berg[1], Chantal Donnelly[1], and David Gustafsson[1]

[1]Hydrology Research Unit, Swedish Meteorological and Hydrological Institute, Folkborgsvägen 17, 601 76 Norrköping, Sweden

*Correspondence to:* Peter Berg (peter.berg@smhi.se)

**Abstract.** Extending climatological forcing data to current and real-time forcing is a necessary task for hydrological forecasting. While such data are often readily available nationally, it is harder to find fit-for-purpose global data sets that span long climatological periods through to near real-time. Hydrological simulations are generally sensitive to bias in the meteorological forcing data, especially relative to the data used for the calibration of the model. The lack of high quality daily resolution data at a global scale has previously been solved by adjusting re-analysis data with global gridded observations. However, existing data sets of this type have been produced for a fixed past time period determined by the main global observational data sets. Long delays between updates of these data sets leaves a data gap between the present day and the end of the data set. Further, hydrological forecasts require initialisations of the current state of the snow, soil, lake (and sometimes river) storage. This is normally conceived by forcing the model with observed meteorological conditions for an extended spin-up period, typically at a daily time step, to calculate the initial state. Here, we present and evaluate a method named HydroGFD (Hydrological Global Forcing Data) to combine different data sets in order to produce near real-time updated hydrological forcing data of temperature and precipitation that are compatible with the products covering the climatological period. HydroGFD resembles the already established WFDEI method (Weedon et al., 2014) closely, but uses updated climatological observations, and for the near real-time it uses interim products that apply similar methods. This allows HydroGFD to produce updated forcing data including the previous calendar month around the $10^{th}$ of each month. We present the HydroGFD method and therewith produced data sets, which are evaluated against global data sets, as well as with hydrological simulations with the HYPE model over Europe and the Arctic regions. We show that HydroGFD performs similarly to WFDEI and that the updated period significantly reduces the bias of the reanalysis data. For real-time updates until the current day, extending HydroGFD with operational meteorological forecasts, a large drift is present in the hydrological simulations due to the bias of the meteorological forecasting model.

## 1 Introduction

Large scale hydrological models at global or continental scales require meteorological forcing data at, typically, daily time resolution. There is a lack of data with high quality and consistency between variables at such scales, however, data at coarser monthly scales are more prominent. Reanalysis data fulfill the spatial and temporal consistency, but suffer from bias that limits their use for hydrological simulations. Current data sets that merge reanalysis and coarser observations bridge the data gap, but are mostly only episodically updated (Sheffield et al., 2006; Weedon et al., 2011, 2014; Beck et al., 2016).

The degree to which the skill of a hydrological forecast is sensitive to the initial hydrological conditions on one hand, and the meteorological forcing in the forecast period on the other hand, depends on factors such as the hydro-meteorological regime of the catchment and the memory of the hydrological system. The hydrological skill sensitivity to the initial state and/or the meteorological forecast varies as a function of the season, which have been shown for both seasonal and short term forecasts (Li et al., 2009; Shukla and Lettenmaier, 2011; Paiva et al., 2012; Demirel et al., 2013; Pechlivanidis et al., 2014). In most cases, however, hydrological forecast models are initialized by hindcast simulations covering some period before the forecast issue date, for which appropriate meteorological forcing data are needed.

Climatological hydrological simulations require consistent forcing data for a long period, which can be problematic with gauge based data sets if the gauge location and the network density are very different between the observed variables. Observational data sets with global coverage are sparse regarding data with at least daily resolution, but there are exceptions such as the Climate Prediction Center's (CPC) products for temperature (CPCtemp, 2017) and precipitation (Chen et al., 2008). There are also several promising satellite based products, such as the TRMM (Huffman et al., 2009b) and GPM missions, although satellite data require adjusments to ground truth observations. The negative aspects of the above data sets are problems with spatial coverage, by non-sampled (polar) regions for the satellite data and lack of gauges in parts of the world for gauge based data as the gauge density becomes even more important at the daily time scale.

Operational models working on a global scale have found ways to work with sparse observations. The Global Flood Awareness System (GloFAS) uses the ERA-Interim reanalysis (Dee et al., 2011), with precipitation adjusted using data from GPCP (Huffman et al., 2009a) at a monthly time-scale (Alfieri et al., 2013; Hirpa et al., 2016). Another global scale model system is the Global Flood Forecasting Information System (GLOFFIS), where the meteorological forcing data is derived from several sources, such as gauge measurements, CPC-Unified gridded precipitation (Chen et al., 2008) and the ECMWF control forecast (Emerton et al., 2016).

Earlier methods (Sheffield et al., 2006; Weedon et al., 2011, 2014; Beck et al., 2016) have merged information from a re-analysis with temporally coarser observational data, to produce new data sets that inherit the temporal resolution of the re-analysis with the average properties of the observations. With these methods, long periods of internally consistent daily or sub-daily resolution and global coverage become available for, e.g., large scale hydrological simulations. The various methods have applied different re-analysis data sets and observational records, and therefore differ in their final result. The more simple method is that of Weedon et al. (2014), where mainly single data sets are applied globally for the adjustment of each variable. Although this leaves the method highly dependent on the quality and availability of few data sets, it makes the method less affected by temporal and spatial inconsistencies between periods and regions. An issue with relying on gridded observational data sets is that such data are often updated episodically, and with several months or even years of delay before they are updated. This can be an issue for global or continental hydrological forecasting where up-to-date information is important, thus requiring a continuous updating of the forcing data while retaining a consistent climatology.

Here, we present the HydroGFD method for producing adjusted meteorological forcing data sets for a near global domain.The novelty in the production of the data sets is the combination of reanalysis and operational global model input, as well as the combination of various observational data sources to fill the gap between the present and the end of the climatological

**Table 1.** Table of meteorological forcing data used in the analyses and hydrological simulations. The data sets are described in Tab.2.

| Abbreviation | Atm. model | Precipitation | Wet days | Temperature | Period |
|---|---|---|---|---|---|
| GFDCL | EI | GPCC7 | CRU | CRU | 1979–2013 |
| GFDEI | EI | GPCC-Monitor | GPCC-FG daily | GHCN-CAMS | 2010–(t-3 months) |
| GFDOD | OD | GPCC-FG monthly | GPCC-FG daily | GHCN-CAMS | 2010–(t-1 month) |
| OD | OD | NA | NA | NA | 2010–t |

products. We evaluate the updating procedure to the climatological data by direct comparison of the meteorological data, as
well as by employing a hydrological model to evaluate the data sets. The main motivation for creating the data set is to update
climatological simulations, but also to improve the initialisation for hydrological forecasting at large scales or in data sparse
regions where dense observational data are not available for initialisation. We present evaluation of two such applications for
the Arctic and European set-ups of the hydrological models E-HYPE and Arctic-HYPE.

## 2  Methods and Data

The HydroGFD method is currently intended to be a substitute and extension of precipitation and temperature from the WFDEI
method (Weedon et al., 2011), which is currently used in many hydrological simulations with HYPE (Lindström et al., 2010)
and other hydrological models.
We are therefore mimicking the WFDEI set-up closely, however, with some necessary differences due to updates of the
meteorological observations since the first appearance of WFDEI. The HydroGFD data set is currently limited to precipitation
and temperature at three and six hourly intervals, wheras WFDEI produces several additional variables (Weedon et al., 2011).
The basic method is to construct monthly mean adjustment factors per calendar month for each variable and to adjust every
time step during the month with that factor. For temperature, the adjustment factor is produced by subtracting the monthly mean
reanalysis from the observations, and adding this to every time step of the reanalysis. For precipitation, a first step of adjusting
the number of wet days is performed. The underlying assumption is that the reanalysis model produces excessive light rainfall
(drizzle). Days with the least amount of rainfall that are in excess to the observed rainy days are set to zero. In a second step,
the ratio between the monthly mean observations and the reanalysis data is calculated and used to scale the reanalysis data.
The HydroGFD system has been applied to produce the main climatological dataset called GFDCL, which is a methodolog-
ical equivalent to the WFDEI (Weedon et al., 2011) dataset except for updated climatological observations (see Tab. 1) and
differences in the implementation. GFDCL, like WFDEI, is based on the ERA-Interim (EI) reanalysis but is coded so that EI
can be interchanged with other reanalyses. Precipitation is corrected for wet day bias compared to CRUts3.22 wet day informa-
tion, and scaled with monthly precipitation from GPCC7 (see Tab. 2). Temperature was corrected additively with CRU monthly
mean temperature. The GFDCL data set is restricted to the time period 1979–2013, due to the start of the EI reanalysis period,
and by the end of the GPCC7 (Schneider et al., 2014) observational data set. The main difference between GFDCL and WFDEI
arises from the treatment of under-catch, i.e. the rainfall likely not captured by the rain gauges due to turbulence around the

**Table 2.** Table of model and data sources used in the analyses.

| Data set | Variables | Resolution | Period | Reference |
|---|---|---|---|---|
| ERA-Interim (EI) | T, P | ~0.8° | 1979–(t-3 months) | Dee et al. (2011) |
| ECMWF-OD (OD) | T, P | ~0.22° | 2010–present | |
| CRUts3.22 (CRU) | T, P, wet-day* | 0.5° | 1901–2013 | Harris and Jones (2014) |
| GPCC7 | P | 0.5° | 1901–2013 | Schneider et al. (2015b) |
| GPCC-Monitor(v5) | P | 1.0° | 1982–(t-2 months) | Schneider et al. (2015a) |
| GPCC-FG | P**, wet-day*** | 1.0° | 2009–(t-1 month) | Ziese et al. (2011); Schamm et al. (2013) |
| GHCN-CAMS | T | 0.5° | 1948–(t-1 month) | Fan and Van den Dool (2008) |
| WFDEI | P****, T***** | 0.5degree | 1979-2013(6) | Weedon et al. (2011) |

\* Gridded from SYNOP stations. ** Using the GPCC First guess monthly product

*** Derived from daily time-step information from the GPCC first guess daily product.

**** Using different versions of GPCC until 2013, also a version using CRU until 2016.

***** Using different versions of CRU until 2016.

gauge. WFDEI applied the Adam and Lettenmaier (2003) under-catch correction to the GPCC5 and GPCC6 data sets. With
GPCC7, under-catch correction is already included in the data set, and need not be applied in the HydroGFD methodology.
However, for GPCC7, the under-catch correction was based on Legates and Willmott (1990), but reduced by 15% to better fit
with their own estimates (Schneider et al., 2014). Adam and Lettenmaier (2003) compared their method with that of Legates
and Willmott (1990) and found the latter to lead to too low precipitation amount by about 5–30%, and differences in the annual
cycle of the correction factors. There is clearly a large controversy on this topic. We therefore expect differences between
GFDCL and WFDEI in both annual totals and in the annual cycle.
The main issue tackled here is how to implement the WFDEI methodology forward in time as GPCC7 becomes unavailable,
or when EI becomes unavailable. We propose two flavours of HydroGFD to extend the period past year 2013 (see Tab. 1 for
data sets and references):

1. GFDEI consists of the EI data set with precipitation scaled by the GPCC monitoring data set and wet day adjusted
   according to the GPCC first guess daily product. Temperature is adjusted with the GHCN-CAMS data set.

2. GFDOD consists of the ECMWF deterministic forecast, which differs from EI by mainly the model version and the
   assimilated data. Precipitation is scaled by the GPCC first guess monthly data set and wet day adjustments according to
   the GPCC first guess daily product. Temperature is adjusted with GHCN-CAMS data.

GFDEI fills the gap between the end of GFDCL in 2013 until the latest available EI data, i.e. until about three months ago.
For the last two months, GFDOD is used to fill the gap. The necessary datasets are all available for download around the $10^{th}$
in each month. Fig. 1 shows a schematic for how the forcing data is used to update hydrological models to today's date. For
example, to update a model to the $9^{th}$ of May, the model is forced with GFDEI until the $31^{st}$ of January, GFDOD until $31^{st}$ of

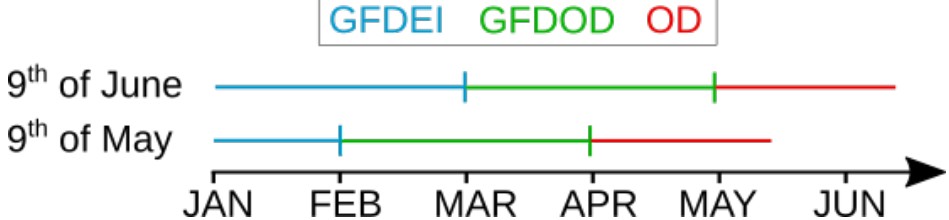

**Figure 1.** Schematic of the updating procedure. The HydroGFD data are continuously updated with GFDEI as long as EI data are available. The intermediary data set GFDOD fills up the time series as long as GPCC data are available, and then continues with uncorrected OD data. Because the previous month becomes updated on the $10^{th}$ of each month, the $9^{th}$ is the day with the longest period of OD driving data. The next month, GFDEI is extended one month, and the GFDOD data are updated for the new month.

March then OD until the $9^{th}$ of May. This gives a period of 40 days with unadjusted OD data. However, to update the model to the $10^{th}$ of May, because the GPCC monitoring product becomes available on the $10^{th}$ of the month (at latest) all data shifts one calendar month and require a shorter period of OD data (unadjusted data). In a hydrological forecasting context, the simulations are updated from the GFDEI data, which is the continuous extension of GFDCL, and the GFDOD and OD parts are re-run after each update to determine the new initial conditions.

Because the observational data sets only provide information over land areas, the HydroGFD system only produces adjustments where data are available, and retains the original reanalysis, or deterministic forecast, when no data is available. One notable exception is Antarctica, which is not covered by the observational data sets, and is therefore not adjusted at any step of the updating procedure.

## 2.1 HYPE model

The HYPE (Hydrological Predictions for the Environment) model is a process based hydrological model developed for high-resolution multi-basin applications, which has been applied at various spatial scales (from tens to million square kilometres) and hydro-climatological conditions (Lindström et al., 2010; Strömqvist et al., 2012; Arheimer et al., 2012; Andersson et al., 2015; Gelfan et al., 2017). The model is based on a semi-distributed approach where the hydrological system is represented by a network of sub-basins, which are further divided into classes that can be selected to represent combinations of soil-type and land-cover or elevation zones. The water balance and runoff from each sub-class is calculated taking into account processes such as snow and glacier accumulation and melt, infiltration, evapotranspiration, surface runoff, tile drainage and groundwater recharge and runoff. The runoff from the land classes is further routed through the network of lakes and rivers represented by the sub-basin delineation. The model is used for research and operational purposes, to provide information for, for instance, flood and hydro power reservoir inflow forecasting, river discharge and nutrient loads to the ocean, as well as assessment of climate change impact on hydrological systems.

To evaluate the real usefulness of the HydroGFD data in continental (and by extension global) hydrological forecasting, the HydroGFD data was tested in two continental scale applications of HYPE. For Europe, the E-HYPE v3.2 (Hundecha et al.,

2016) hydrological model was calibrated with GFDCL and employed to evaluate the updating versions of HydroGFD. The
simulation domain ranges from wet Arctic, wet maritime to dry Mediterranean climatic conditions. The E-HYPE model has
been shown to reproduce well the spatial and temporal variability in hydrological processes across Europe (Donnelly et al.,
2016; Hundecha et al., 2016), and has been identified as a useful model for continental scale forecasting (Emerton et al.,
2016). E-HYPE takes daily mean precipitation and temperature as input. Potential evapotranspiration is estimated from daily
mean temperature and extraterrestrial radiation estimated separately for each sub-basin location and day of the year using the
modified Jensen-Haise/McGuiness model following Oudin et al. (2005). For each sub-basin, air temperature and precipitation
is taken from the nearest grid point. Temperature is further corrected with a constant lapse rate (-0.65 $^{\circ}C/100\ m$) for the
difference between the mean sub-basin elevation and the corresponding elevation of the grid point. Elevation correction of
precipitation is also possible in the HYPE model, but it is not used in E-HYPE.

11       For the Arctic, we use the Arctic-HYPE model v3.0 (Andersson et al., 2015; Gelfan et al., 2017) that covers the land area

draining into the Arctic Ocean (excluding Greenland). The model domain is 23 million $km^2$ divided into 32599 sub-basins
with an average size of 715 $km^2$. The Arctic region is characterized by numerous lakes of various size (5% areal fraction) and
glaciers (about 50% of the glaciated area outside the Greenland and Antarctica Ice sheets, mainly on islands in the Canadian
Arctic archipelago, Svalbard, and Russian Arctic islands) (Dyurgerov and Meier, 1997; Meier and Bahr, 1996). To take into
account the long turnover times of larger lakes in the domain (for instance Lake Baikal) and the on-going decline in glacier
volume, the Arctic-HYPE model was initialized using an initial spin-up period for the period 1961–2010 using the WFD data
(Weedon et al., 2011) with a simplified correction of precipitation versus GPCC7 on a monthly basis, to be consistent with
the GFDCL data, and extended using GFDCL for the period 1979-2013. As for E-HYPE, Arctic-HYPE is forced by daily
mean precipitation and temperature, but in contrast to E-HYPE, potential evapotranspiration is calculated using the Priestley-
Taylor equation assuming it to be more representative for the wide range of climatic conditions in the Arctic-HYPE domain.
The Priestley-Taylor equation requires solar radiation and relative humidity, which was estimated using the minimum and
maximum daily temperatures as additional input variables, following the recommended procedures by Allen et al. (1998).

24       Both E-HYPE and Arctic-HYPE models have been parametrized and calibrated with similar step-wise approaches involv-

ing first of all sub-basin delineation based on globally available digital elevation data (USGS HydroSHEDS and Hydro1K).
Secondly, classification into selected land-use and soil type classes based on land cover and soil data such as the ESA CCI
Land cover or CORINE, and HWSD respectively. Thirdly, model parameters governing water balance processes in ice/snow,
soil, lakes and rivers were thereafter calibrated in an iterative procedure using river discharge data from the Global Runoff
Data Center (GRDC), as well as data on internal water balance components such as snow (ESA GlobSnow and Former Soviet
Union Snow course data), glaciers (glacier area and mass balance data from ESA CCI Glacier and the World Global Monitoring
Service), and evapotranspiration (fluxtower data from FluxNet and MODIS Evaporation products).

32       For the evaluation simulations with HydroGFD products, the models are run once per month from $9^{th}$ of May 2010 to $9^{th}$

of December 2013, to recreate a 130 day initialization simulation for each run, ending on the given date. This is the longest
possible initialization step, as the meteorological forcing data are updated at the $10^{th}$, for which the initializations would
advance one calendar month (Fig. 1). The first simulation starts from a saved state of the GFDCL simulation in January 2010,

and each subsequent run is initialised from a starting state saved from the GFDEI portion of the previous simulation; making the GFDEI simulation continuous in time. A total of 44 simulations are made with each hydrological model. The simulations are then compared with a climatology simulated using GFDCL forcing for each region for the same period 2010–2013 to evaluate the change in simulated hydrology as a result of the changing forcing data products.

# 3   Results

We begin with evaluating the GFDCL data set, as well as comparing differences between the various HydroGFD versions. Thereafter, we present analysis of hydrological simulations for Europe and the Arctic.

## 3.1   Meteorological evaluation

**Climatology 1979–2013:** GFDCL is directly comparable to the WFDEI data set due to the very similar method, but will differ due to different underlying data, and handling of precipitation under-catch. Because WFDEI was on several occasions evaluated against flux tower measurements across the globe (Weedon et al., 2011, 2014; Beck et al., 2016), we do not repeat such evaluation for GFDCL here, and compare instead to the WFDEI and other data sets.

The baseline reanalysis data set EI has both wetter and drier regions compared to GPCC7, with biases towards $\pm 100\%$ over large regions (Fig. 2b). Overall, the wetter regions are predominant. Here, we note especially the wet bias throughout the Arctic (excluding Greenland), and mainly slightly wet bias in continental Europe. Corrections with GFDCL reproduces GPCC7 well (Fig. 2c), as expected per definition of the method. There are some isolated patches with underestimated precipitation, mainly in the dry regions of the Sahara desert and southern Arabic Peninsula, which appears because no scaling is possible for single months with a complete lack of precipitation in EI at these locations. In contrast to GFDCL, WFDEI has a general wet bias when compared to GPCC7 (Fig. 2d). The wet bias is explained mainly by stronger under-catch corrections included in WFDEI, as explained in Section 2.

Temperature bias in EI ranges mainly between $\pm 1\ ^{\circ}C$ for most land areas (Fig. 2f), but there are regions with considerable bias. There is a mostly warm bias of partly several degrees Celsius in the Arctic regions. Europe has a low bias, except for Scandinavia, which shows a warm bias. Both GFDCL (Fig. 2g) and WFDEI (Fig. 2h) correct the bias per definition, and are both indistinguishable at the $0.2\ ^{\circ}C$ accuracy of the color legend, even though different versions of CRU were employed (GFDCL: CRUts3.22; and WFDEI:CRUts3.1 for 1979–2009, CRUts3.21 for 2010–2012, and CRUts3.23 for 2013).

In summary, GFDCL is methodologically similar to WFDEI and differences in the results are mainly due to the different precipitation source used.

**Evaluation of the updating method (2010–2013):** To evaluate the updating method of the GFDEI and GFDOD datasets, we investigate differences in bias for the period 2010–2013 when all data sources are available (see Tab. 2). The only methodological difference between GFDEI/OD and GFDCL is the calculation of the number of wet days in a month. Whereas the latter uses gridded station measurements of the number of wet days from CRU, the former data sets have the number of wet days calculated from the GPCC-FG daily product as the number of days in a month with precipitation larger than or equal to

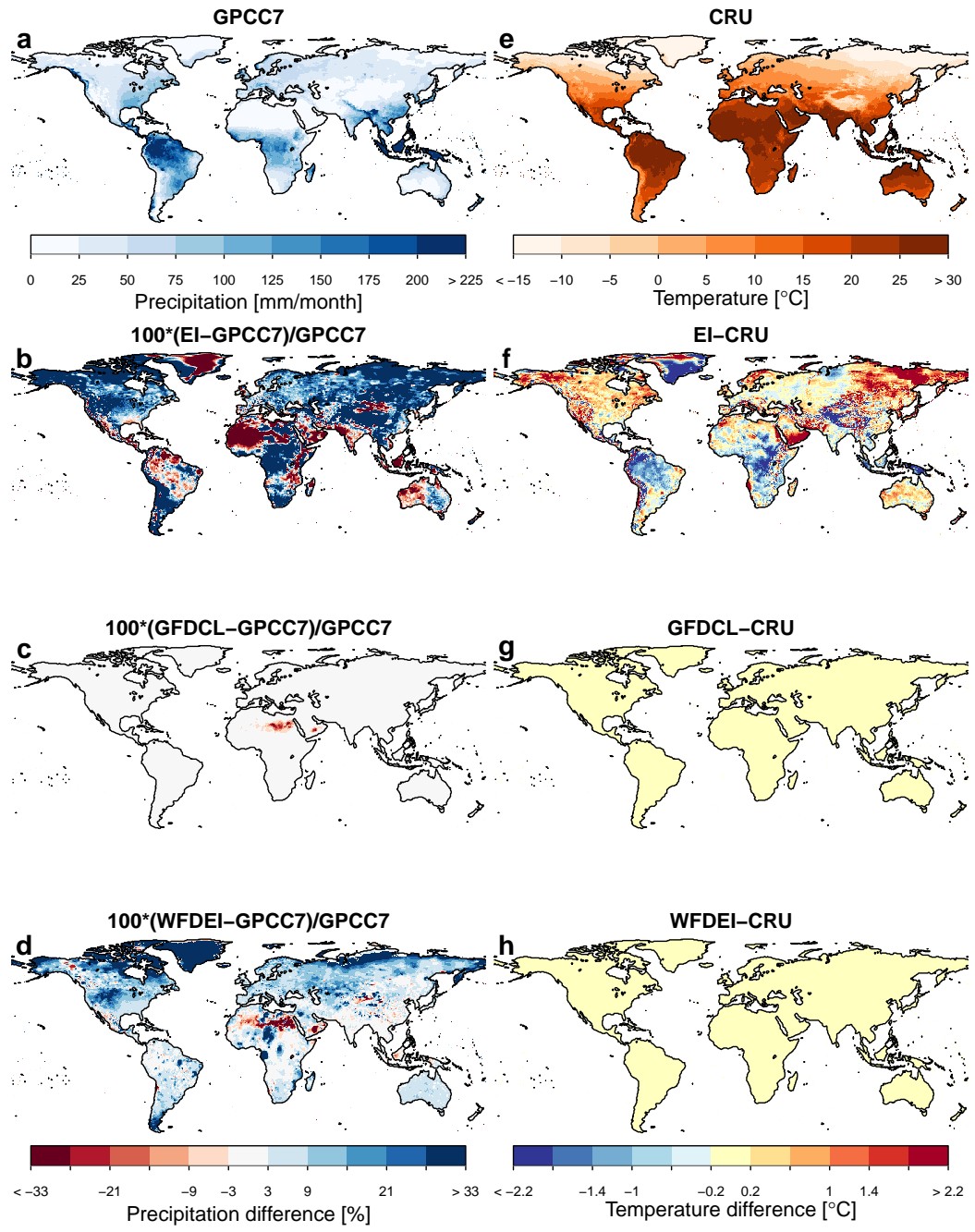

**Figure 2.** Climatology of (a) precipitation from GPCC7, and (e) temperature for CRU. Relative difference in climatological precipitation from GPCC7 for (b) EI, (c) GFDCL and (d) WFDEI. Absolute difference in climatological temperature from CRU for (f) EI, (g) GFDCL and (h) WFDEI.

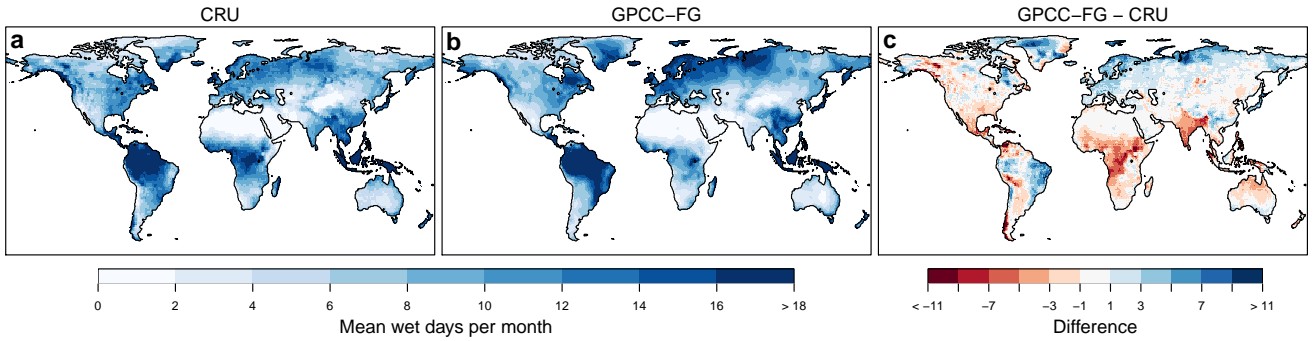

**Figure 3.** Comparison of the number of wet days provided by (a) the CRU data set, compared to those derived from (b) GPCC-FG, and (c) the difference between the two for the period 2010–2013.

1 mm/day. Fig. 3 presents the period average number of wet days in a month for CRU and GPCC-FG. The two methods to calculate wet days differ significantly for Europe and especially the Arctic part of Scandinavia and western Russia, where the updating method overestimates the number of wet days. The updating method also produces underestimations in Africa, Latin America and the Andes. An interesting difference is markedly confined within the political borders of India, which implies a difference in the observations entering either CRU or GPCC-FG, and could be an artefact of a higher station density in that region compared to surrounding regions or a different threshold used for the wet-day definiton.

Fig. 4 shows the bias between the different data sets used here, such that the data set given at the top of the plot is compared with that named to the left of each row. In the first row (Fig. 4a–d), all data sets are compared to GPCC7. Clearly, GPCC-Monitor and GPCC-FG both underestimate precipitation for most parts of the globe compared to GPCC7. This is partly due to the lack of under-catch correction, but differences may also result from lower station density, as not all stations are available in real-time. The latter effect can be seen in the different bias patterns for GPCC-Monitor and GPCC-FG (Fig. 4a and b, respectively), and also in the difference between GPCC-Monitor and GPCC-FG (Fig. 4e). The extension of the GFDCL data set is mainly through the GFDEI product, which is adjusted by GPCC-Monitor, and the GFDOD product is mainly used as an interim measure to bridge the data gap for initializations of forecasts. GFDEI has a similar spatial structure as GPCC7, with some marked regional differences, but a general reduction of a few percent in total precipitation is seen. EI has a similar bias as for the climatological period (compare Fig. 4c and Fig. 2b). The bias of GPCC-Monitor shrinks in significance when compared to that of EI, which means that the extension of GFDCL with GFDEI is indeed relevant when extending the climatological data set for, e.g., hydrological applications.

OD has a similar bias as EI when compared to GPCC7 (Fig. 4d), however, also clear differences although of lower magnitude appear in a direct comparison of OD and EI (Fig. 4j). The main differences are confined to the tropical regions, however, the bias of OD is much more prevalent than that of GPCC-FG, which indicates value in the interim GFDOD product. GFDEI and GFDOD retains the average bias of the GPCC-Monitor and GPCC-FG products, per definition (not shown).

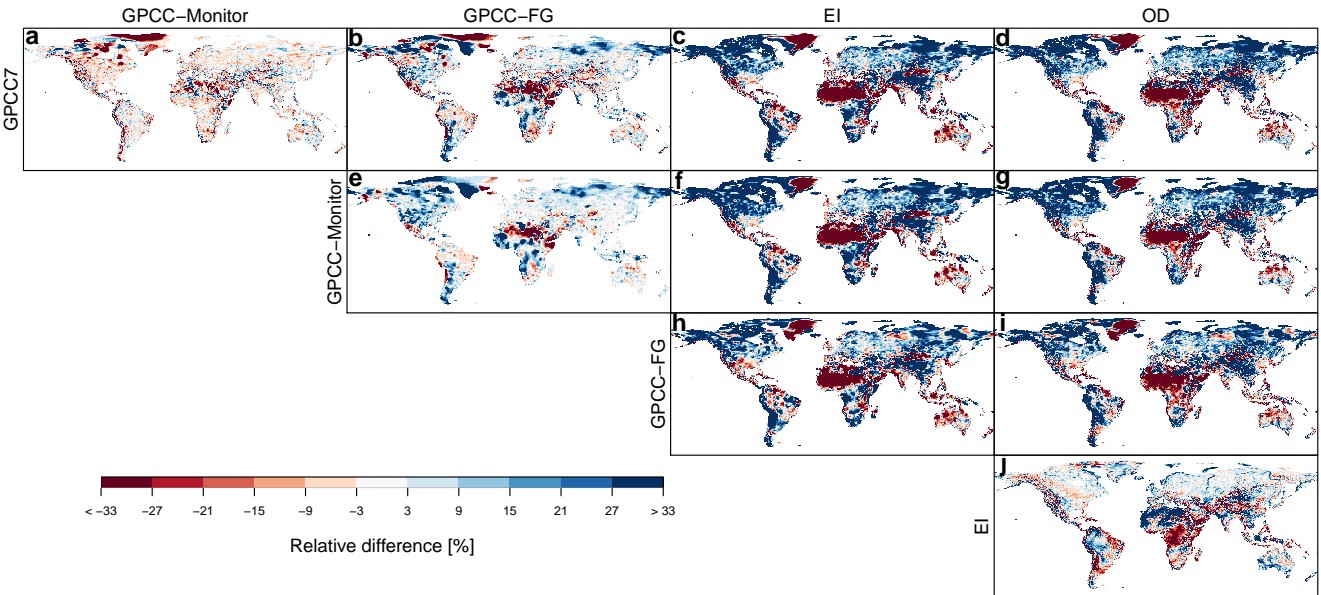

**Figure 4.** Relative difference of mean monthly precipitation between different data sources and (a–d) GPCC7, (e–g) GPCC-Monitor, (h–i) GPCC-FG, and (j) EI.

Temperatures are compared between the data sets GHCN-CAMS, EI and OD toward CRU (not shown). The main differences
are in the Arctic, especially for Greenland, and for various mountain ranges and coastal areas, with magnitudes of several
degrees Celsius. EI and OD have similar bias for most of the globe, although OD has a larger warm bias in the Arctic and
northern Europe.
**3.2   Hydrological evaluation**
The effect of the interim products on simulated hydrology in Europe and the Arctic are evaluated using the E-HYPE and
Arctic-HYPE continental hydrological models. The resulting bias at the end of OD simulation is indicative of the potential
bias in initial conditions for a hydrological forecast made using the HydroGFD procedure. First, a climatological simulation
driven by GFDCL is carried out for the years 2010–2013, starting from a saved model state the $10^{th}$ of January 2010. Second,
a set of simulations separated by one calendar month was carried out for the period $10^{th}$ of May 2010 until $10^{th}$ of November
2013. Each of the simulations start from GFDEI for the first month, continue with GFDOD for two months, and then OD for
one month and ten days (see Fig. 1). The model state of the last day of the GFDEI simulation is saved and used for the initial
state of the next month's GFDEI simulation. When nothing else is stated, the evaluation is performed with day one at the first
day of the GFDEI until the last day of the simulation, which is approximately day 130. In the figures we mark with colours as
in Fig. 1 the different forcing data periods approximated by 30 day months to indicate which data set was used.

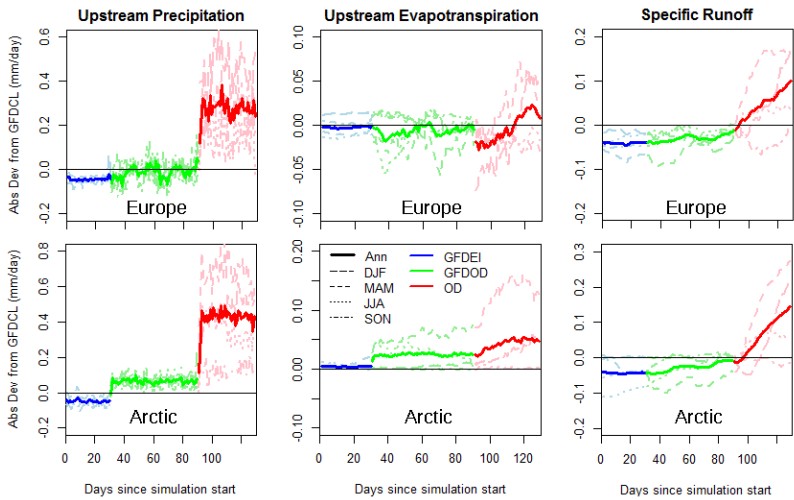

**Figure 5.** Upstream precipitation, evapotranspiration and specific runoff averaged over all catchments and shown for all forecast times as well as per season for (top) E-HYPE and (bottom) Arctic-HYPE. All runs are presented as deviations from the GFDCL forced simulation.

The impact of the differences in the GFDEI, GFDOD and OD data sets compared to the reference GFDCL simulation are
shown as an average across the respective simulation domains in Fig. 5. The specific runoff shows lower values for GFDEI
and GFDOD compared to GFDCL for both domains. Clearly, the main determining factor for the differences arise from the
differences in upstream precipitation from the first 30 days with GFDEI. Even though GFDOD has less of a precipitation offset
from GFDCL, and for the Arctic even a positive difference, the GFDEI offset causes a slow drift in runoff toward the new
conditions of GFDOD, and therefore a remaining negative offset for about the first 90–100 days. Upstream evapotranspiration
shows a low offset from GFDCL for GFDEI, which shows that the GHCN-CAMS and CRUts3.22 data sets are similar for these
two domains. However, although the same data set is used for GFDOD, there is a larger offset for this period. The difference
in upstream evapotranspiration offsets between the two model domains is most likely a result of the larger (and positive) offset
in upstream precipitation for the GFDOD and OD periods in the Arctic-HYPE domain, rather than the smaller differences
in temperature. OD has a strong wet precipitation bias (particularly in the northern hemisphere; results for the tropics and
southern hemisphere may be different)(Fig. 4d), which is of a much greater magnitude than that of GFDEI. The bias causes
the slow drift of the specific runoff to accelerate around day 90–100, as the model adjusts to the new precipitation average. The
case is similar for both domains. Another striking feature from Fig. 5 is the larger variability for GFDOD and OD, compared
to GFDEI, which is due to differences between EI and OD. This affects the day-to-day variations of the simulations, but not
the total water balance.
Fig. 5 shows also results per season. For both Europe and the Arctic, precipitation and runoff biases are largest for the OD
forced period in DJF and MAM, and relatively minor in JJA and SON. Seen as a continental mean, there is little variation in
the biases between individual years, meaning that the results are robust in time (not shown).

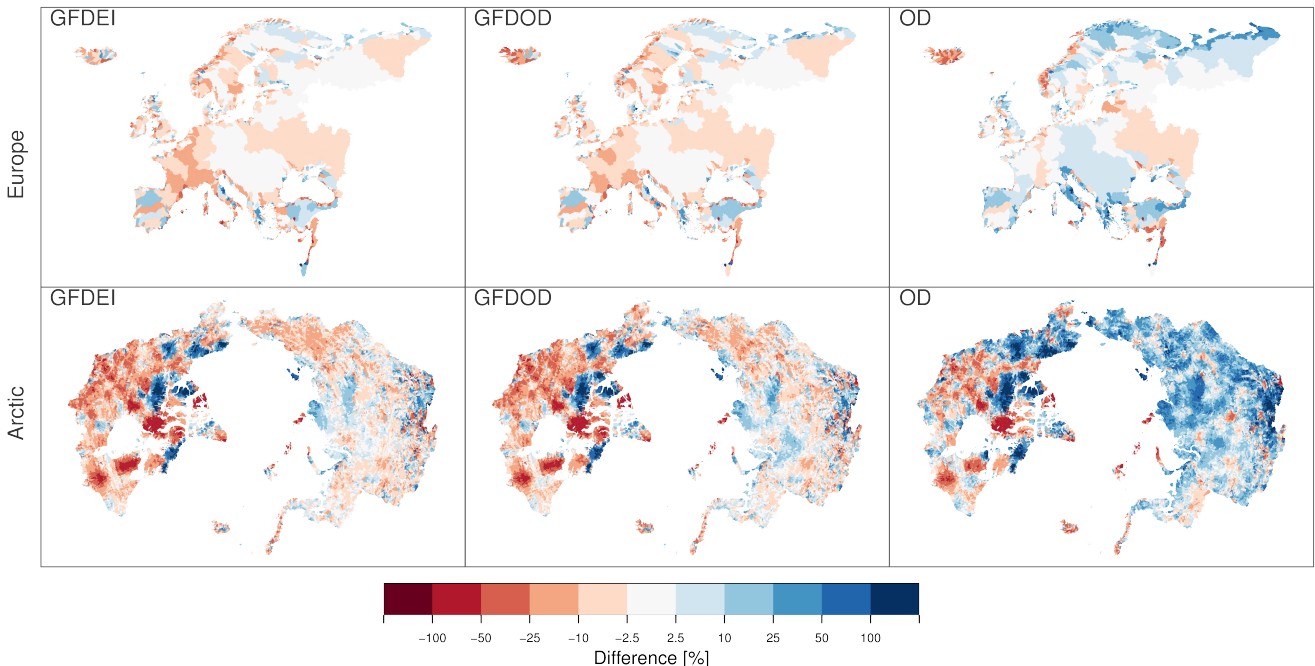

**Figure 6.** Relative upstream specific runoff difference from GFDCL for each catchment of (top) E-HYPE and (bottom) Arctic-HYPE, with the different data sets (right to left) GFDEI, GFDOD and OD.

Fig. 6 shows a spatial view of the average upstream runoff difference from the GFDCL simulation for each domain. In the resolution of the colour scales, there are only small differences between GFDEI and GFDOD. The offsets from GFDCL are mainly within ±20% for Europe, but much stronger local offsets are seen in the Arctic domain. The Arctic is a more sensitive region to differences in the station density behind the gridded observational data sets, as there are fewer stations to begin with. This fact plays a large role in shaping the offsets seen here. The OD period is, as expected, wetter for most of the domains, but more clearly so for the Arctic domain.

A selection of in-situ observations from gauging stations with available data from at least two of the four simulated years was used to analyse how the model performance against observed discharge varies using the climatological forcing and different interim data sets. Performance criteria of the models for each of the gauges are presented for each data set in comparison to GFDCL in Fig. 7. Since GFDCL is always the reference, the results for each gauge lines up vertically in the figure. The two domains show similar results, and we therefore describe the results in a general sense. The bias follows the patterns described above, with lower values for GFDEI and GFDOD, while OD has higher values. Whether there is a positive or negative bias is determined by the initial bias of the GFDCL simulation. NSE and Pearson correlation (r) are not showing any clear structure, but remain reasonable for most of the simulations. The variance is consistently higher for the OD simulation as also noted above.

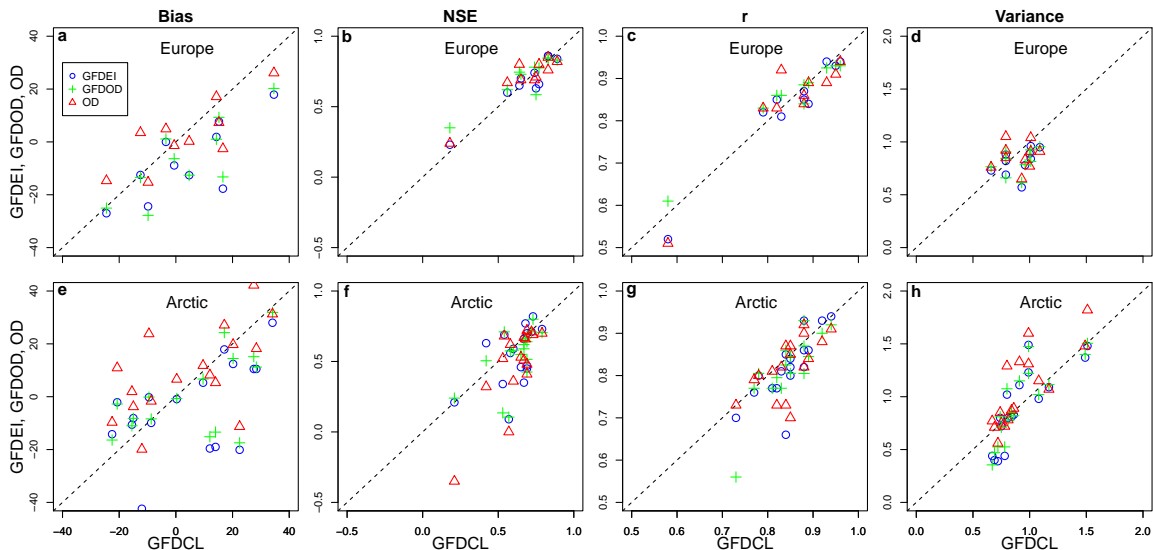

**Figure 7.** River discharge model performance measures: bias (relative volume error in %), Nash-Suthcliff Efficiency (NSE), Pearson correlation (r), and ratio of simulated and observed variance for a selection of grid points in (top) Europe, and (bottom) Arctic. The performance of GFDEI, GFDOD, and OD (y-axis) is compared to GFDCL (x-axis) in scatter diagrams.

In summary, the domain average deviations from GFDCL shows that the updating procedure adds value to the simulations
by keeping the precipitation and temperature climate closer to the GFDCL data set when compared to the alternative of using
uncorrected data (e.g. OD). The extension of GFDCL with GFDEI has only minor effects on the long term hydrology. However,
for forecast initializations, the inevitable switch to OD data closer to the "current" date, i.e. the day to issue a forecast, there
is a strong drift due to the wet bias of OD in the northern hemisphere regions evaluated here. The drift continues throughout
the OD period, which means that the initial drift a forecast is subjected to is dependent on the day of the forecast. The drift is
largest for forecasts issued just before the $10^{th}$ and lowest just after. This warrants future development to look for a method to
adjust the deterministic forecast data (OD). In highly seasonal regions with little interannual variability, OD could be adjusted
with the monthly climatological mean precipitation and temperature; however, it should be investigated whether this worsens
simulations in regions with high interannual variability. Such a correction could also be used within the forecasting period;
however is reserved as the subject of future study.
**4   Conclusions**
We present and evaluate a new data set called HydroGFD, which consists of several interim products to fill the gap between
available climatological and forecasted data. The main product, GFDCL, is the methodological equivalent to the already well
established WFDEI (Weedon et al., 2014), although with updated observational data sets. To extend the data set beyond year

2013, when e.g. the GPCC7 data set ends, adjustments are performed with regularly updated data sets. This is performed with the GFDEI product until the latest update of EI, which is with about a three month delay. For near real-time updates, GFDOD makes use of the ECMWF deterministic model with similar data sets for adjustments as for GFDEI. GFDOD is available until the end of the previous month from around the $10^{th}$ of the current month.

GFDCL is found to be a much similar product to WFDEI, but with a more consistent data set. The introduced under-catch corrections in the precipitation data set GPCC7 differ from that assumed in WFDEI, which leads to generally lower amounts in GFDCL. Temperature is very similar.

The updates in GFDEI beyond 2013, are evaluated for an overlapping period (2010–2013). GFDEI is found to have slightly lower precipitation amounts, and spatially somewhat different temperatures. However, the differences to GFDCL shrinks in comparison to the bias of EI which has bias of often an order of magnitude higher.

When EI is not available, the OD model is employed and the precipitation data source changes from GPCC-Monitor to GPCC-FG. The change in data source has the largest impact, with several geographical differences which impact on the GFDOD product. As an interim product until the next update, GFDOD reduces the bias of OD (which is similar to that of EI) to levels similar to GFDEI.

Initializations of hydrological simulations for forecasting purposes are investigated for GFDOD, extended by the non-corrected OD until the day before the next update of GFDOD. It is found that the strong bias of OD, especially for precipitation, causes a severe drift of the hydrological model away from the GFDOD climatology. The results are similar for both the domains investigated, i.e. Europe and the Arctic region. Some measure to reduce the induced drift due to bias of OD would be necessary for reliable forecasts. Further, as HydroGFD data are updated, it is necessary to re-run the hydrological model from the last update of EI, i.e. for the last three months. The effect of the updating procedure will be that the forecast just after the update, will not be consistent with the one from the day before due to the change in the last few months and the initial state at the time of the forecast. Analysis of the forecasts was not part of the current study.

HydroGFD is currently applied for forecasts with HYPE models in the Niger river basin (http://hypeweb.smhi.se/nigerhype/) which is evaluated in (Andersson et al., 2017), the Arctic (http://hypeweb.smhi.se/arctichype/), as well as for seasonal forecasts in a concept study for Copernicus Climate Change Services available from the sectoral information services at the website http://climate.copernicus.eu/.

The HydroGFD data sets are planned for public release via a web interface on http://hypeweb.smhi.se/. An updated version of HydroGFD using the new reanalysis system ERA-5 and introducing further observational data sets is foreseen during 2018.

## 5  Data availability

The HydroGFD method relies mainly on open data sets, as referenced within the article. ECMWF reanalysis can be accessed via the web portal https://www.ecmwf.int/en/research/climate-reanalysis/era-interim/. The forecasts from ECMWF (here referred to as "OD"), are restricted to member institutes (or other special circumstances, see https://www.ecmwf.int/en/forecasts/accessing-

forecasts), and are therefore not available for public download. However, HydroGFD will shortly appear online on http://hypeweb.smhi.se/.
Hydrological simulations were performed with the open source model HYPE, which can be accessed at http://hypecode.smhi.se/.
*Acknowledgements.* We acknowledge the hard work of building the data sets used within the presented work. This includes the data from the
ERA-Interim, CRU, GPCC and WFDEI as referenced within the paper, as well as GHCN-CAMS (National Center for Atmospheric Research
Staff (Eds). Last modified 08 May 2014. "The Climate Data Guide: GHCN (Global Historical Climatology Network) Related Gridded Prod-
ucts." Retrieved from https://climatedataguide.ucar.edu/climate-data/ghcn-global-historical-climatology-network-related-gridded-products.)
and the ECMWF deterministic forecast system. Further, we acknowledge the inital work of implementing the HydroGFD system at SMHI
by Lisa Bengtsson, Magnus Lindskog and Heiner Körnich, and the work on operationalization by Fredrik Almén.

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
