# Peer review of "Near real-time adjusted reanalysis forcing data for hydrology"

_Hydrology and Earth System Sciences, 2017_

## Referee Comment (RC1) · G.P. Weedon (Referee) · 4 Sep 2017

Berg et al provide a method to produce updates to near "real time" of half-degree spatial resolution daily near-surface temperature and precipitation. These data, called GFD, are designed to allow hydrological modelling that is closer to real time than possible via the episodically released datasets such as the WFDEI. Overall this is a good, clearly written paper which should definitely be published. Below I provide a few minor comments that should be addressed followed by small changes to figures and a few very minor text corrections.

Minor comments: 1) Although the authors target updating beyond the current coverage of GPCCv7 precipitation (2013) using GPCC products, in fact datasets such as

WFDEI already extend to the end of 2015 through the use of the CRU observations of precipitation. Therefore the need for updating is not as severe as implied, especially for Europe where the half-degree resolution GPCC and CRU precipitation totals show good overall matches. Additionally, the post 2013 data (CRU of Rainf_WFDEI_CRU plus Snowf_WFDEI_CRU) could be used within the validation of the new product. The reason why CRU precipitation for 2014 and 2015 has not been utilized should be made clear.

2) The abstract should make it clear that the new product is available at half-degree resolution and for daily precipitation and near-surface air temperature only (other variables provided by the existing datasets, such as downwards shortwave radiation fluxes [suitable for land surface models and global hydrological models] are not provided).

3) The name GFD (Global Forcing Data) is very generic. This name does not convey the fact that only temperature and precipitation are involved nor that the data are updated to near real time. I would suggest a change to something like Current Global Forcing Data to emphasise the value added.

4) The updating methodology is dependent on the availability of ERA Interim products. In the next couple of years ERA Interim will no longer be available and a different reanalysis (ERA-5) will be provided by ECMWF instead. The nature of ERA-5 is such (higher resolution, multiple realizations) that a smooth transition from ERA-Interim to ERA-5 for the GFD is not guaranteed. Some comment on this would be appropriate.

Figure changes: Fig 2 is currently difficult to comprehend. The caption says "Climatological mean (top) precipitation" - the brackets should say "(left)" similarly "and (bottom) temperature" - the brackets should say "(right)". The top row shows absolute means and the colour bars should be provided next to them. The next three rows show differences and the other colour bars should be there. However, the caption says: "the relative difference EI, GFDCL, and WFDEI." This should be changed to "(left) the relative difference in precipitation and (right) absolute difference in temperature." Finally to

[Figure]

be clear what is shown every panel should have its own heading. For example, panel b should be headed with something like: "100%x(EI minus GPCCv7)/GPCCv7".

Fig 6 It is noticeable that the authors have been careful to avoid poor colour pallets that could confuse colour-blind readers. However, it is very hard to distinguish the yellow shades as well as the blue shades for the E-HYPE maps. Can the colour scheme be changed to show clearer gradations?

Fig 7 The labelling is misleading. It is far easier for the reader to understand this figure if every panel has a heading of either "Europe" or "Arctic". Also every Y axis should have the evaluation metric indicated for every panel ("Bias", "NSE", "r", "Variability"). The caption should also spell out NSE, r (is this Pearson's or Spearman's?) and what is meant by "variability" (is this standard deviation or variance?).

Minor text corrections: p4 line 14 The word "data" is plural. A dataset (singular) contains a lot of data (plural). Hence in both places on this line change "is" > "are".

p4 line 14 "On notable" > "One notable".

p5 line 33 and p6 line 1 "Priestly" > "Priestley".

p8 line 5 "method overestimate" > "method overestimates".

p8 line 5 "The updating method also produce" > "The updating method also produces".

p8 line 7 "a difference already in the observations" > "a difference in the observations".

p8 line 15-16 "is mainly used interim to bridge" > "is mainly used as an interim measure to bridge".

p10 line 6 "for the first about 90-100" > "for about the first 90-100".

p10 line 11 "which is on a much larger magnitude" > "which is of a much greater magnitude" [note the "of" not "on"].

[Figure]

326, 2017.

---

## Referee Comment (RC2) · Anonymous Referee #2 · 8 Sep 2017

In this manuscript Berg et al. provide a method to produce "near real time" global forcing data for hydrological models. The methodology is closely based on the methodology used for the WFDEI dataset and extends it with minor modifications to produce "near real time" global forcing data. Overall I think there is an insufficient review of other relevant datasets and methodologies that are clearly linked with this manuscript. In addition, the discussion on the methodology and its advantages and limitations is insufficient. Below I have outlined my major and minor comments. A major review is necessary for this manuscript before it can be published.

Major comments:

1.) In the introduction the authors claim that "forcing data for large scale hydrological models is essentially not available..." – This claim is not correct. In fact there are

numerous, (semi) operational global hydrological models that have solved the problem of forcing data in different ways, for example the Global Flood Awareness System, the Global Flood Monitoring System, and the Global Flood Forecasting System. Please refer to their relevant scientific articles on what type of global forcing data they use to derive hydrologic model initial conditions. In addition, see also the recent article from Hirpa et al. (AMetSoc, 2016) on this topic which should be considered by the authors. Finally, also ERA-5 is now available and produced in "near real time". This should also be included and discussed.

2.) Numerous datasets that are claimed to lack temporal coverage have in fact coverage also of recent years including the MSWEP dataset. The manuscript should reflect the latest status of these datasets.

3.) Furthermore, the authors have omitted completely the TRMM (and now Global Precipitation Measurement) datasets that represent an important source of near-real time precipitation forcing which is clearly the most important variable for the forcing data of hydrological models. Those need to be at least mentioned/referenced with an explanation of why those are not used in this work.

4.) The authors claim that their method is similar to the method used in the WFDEI dataset. Yet, in Fig. 2d the relative difference between WFDEI and GPCC is considerable whereas the relative difference between GFDCL and GPCC is very small. This suggest that the changes introduced by the authors in comparison to the methodology to WFDEI have a significant impact. Instead of claiming that this is simply due to the use of different precipitation sources this should be further investigated and explained.

5.) This manuscript has almost 20 (!) abbreviations. Some of them are spelled out before their first use, others not. Some important ones such as GFDOD1 and GFDOD2 are not properly explained. Even though I am familiar with most of the abbreviations it makes this manuscript very hard to read. The authors should at least include a table with an overview of the most important ones (maybe expand tables 1 and 2) or maybe
add them as an annex.

6.) What is the effect on hydrological simulations when you update with the new observational data on the 10th of each month? This might lead to a significant discrepancy between simulations done on the 9th and then, with the updated dataset on the 10th. Clearly that represents an issue for hydrological forecasting but is not properly discussed by the authors.

7.) Figure 6 d seems to suggest that there is actually less or at least similar bias in the average upstream runoff difference when compared to 6b and 6c. This seems to contradict Figure 5 where the OD period shows the highest absolute difference. Please explain more in detail why this is the case.

8.) The GFD claims to be a global dataset for hydrological models. Yet, the hydrological validation was only performed for catchments in northern latitudes. There is currently no hydrological validation for basins located in tropical climates. The validation should be improved including also basins from these regions.

9.) The manuscript lacks a paragraph on future developments.

10.) The title of the manuscript should be modified and the authors should define in the text what they mean with "near real time". "Near real time" suggests that data is updated within hours or at least days and not monthly.

Minor comments:

- Please add the datasets used for WFDEI to Table 2

- Please add a reference for GHCN-CAMS into the references

- P.10, line 14: last sentence is unclear. Please describe further and rephrase

- What is the difference between GFDOD1 and GFDOD2?

- The nonlinear scale in Fig. 2 and 4 makes it very difficult to look at the results.

[Figure]

Basically everything greater than +-25% is hardly distinguishable. Please choose a different scale or use the one applied in Fig. 6 (and possible also a different color scale)

- In the section on "Meteorological evaluation" the authors write "...and focus instead on comparisons to the WFDEI dataset." However, in the following evaluation you compare the GFDCL mostly with GPCC, EI or OD. Please clarify.

- Does Fig. 2 show the relative difference of EI, GFDCL and WFDEI to GPCC7/CRUts? Please make this more clear.

- Is the precip bias between EI and GPCC7 in line with other studies looking at the precip bias of EI? If yes please add the relevant reference.
* * *

---

## Author Response (AR1)

**Response to reviewer #1:**

**Dear Graham,**
**We appreciate the time taken on reviewing our manuscript, and the comments you have made. Our response to each comment are given in bold font below:**

Minor comments:
1) Although the authors target updating beyond the current coverage of GPCCv7 precipitation (2013) using GPCC products, in fact datasets such as WFDEI already extend to the end of 2015 through the use of the CRU observations of precipitation. Therefore the need for updating is not as severe as implied, especially for Europe where the half-degree resolution GPCC and CRU precipitation totals show good overall matches. Additionally, the post 2013 data (CRU of Rainf_WFDEI_CRU plus Snowf_WFDEI_CRU) could be used within the validation of the new product. The reason why CRU precipitation for 2014 and 2015 has not been utilized should be made clear.

**It is true that the CRU-version of WFDEI is updated more frequently than appears from our manuscript, and we will make statements to emphasise the more freuqent updates. We have not used any data after 2013 because we wanted to have a complete set of data for all data sets, and we believe the WFDEI product based on GPCC data is the better one globally as well as more compatible with the current GFDCL version. Extending the analysis beyond 2013 would therefore introduce more even more complexity in the paper with multiple versions of WFDEI. We will therefore not extend the analysis beyond 2013.**

2) The abstract should make it clear that the new product is available at half-degree resolution and for daily precipitation and near-surface air temperature only (other variables provided by the existing datasets, such as downwards shortwave radiation fluxes [suitable for land surface models and global hydrological models] are not provided).

**We now mention that on several occasions in the revised version.**

3) The name GFD (Global Forcing Data) is very generic. This name does not convey the fact that only temperature and precipitation are involved nor that the data are updated to near real time. I would suggest a change to something like Current Global Forcing Data to emphasise the value added.

**Point well taken, and we have on several occasions considered a different name of the product. It has however been used in many applications already and it would be confusing for current users to re-name it now. We have after some thought made a name change to the overall method to HydroGFD, because hydrological applications is the main aim of this data set. This also separates it more clearly from the WFD data set which could otherwise be confusing.**

4) The updating methodology is dependent on the availability of ERA Interim products. In the next couple of years ERA Interim will no longer be available and a different reanalysis (ERA-5) will be provided by ECMWF instead. The nature of ERA-5 is such (higher resolution, multiple realizations) that a smooth transition from ERA-Interim to ERA-5 for the GFD is not guaranteed. Some comment on this would be appropriate.

**GFD is generic in the sense that it can easily be applied to other re-analysis and observational data sets. We will comment on our plans for switching to ERA-5 once enough data has been released in 2018, and that we will then rebuild the complete data product to make use of the**

**full ERA-5 data.**

Figure changes:
Fig 2 is currently difficult to comprehend. The caption says "Climatological mean (top) precipitation" - the brackets should say "(left)" similarly "and (bottom) temperature" - the brackets should say "(right)". The top row shows absolute means and the colour bars should be provided next to them. The next three rows show differences and the other colour bars should be there. However, the caption says: "the relative difference EI, GFDCL, and WFDEI." This should be changed to "(left) the relative difference in precipitation and (right) absolute difference in temperature." Finally to be clear what is shown every panel should have its own heading. For example, panel b should be headed with something like: "100%x(EI minus GPCCv7)/GPCCv7".

**We appologize for the incorrect information in this figure caption due to a last minute restructuring. We have made changes both to the caption and the figure itself to make everything more clear.**

Fig 6 It is noticeable that the authors have been careful to avoid poor colour pallets that could confuse colour-blind readers. However, it is very hard to distinguish the yellow shades as well as the blue shades for the E-HYPE maps. Can the colour scheme be changed to show clearer gradations?

**We have adapted the colorbrewer palettes and have thereby restricted ourselves to few and clearly distinguishable colors for all plots.**

Fig 7 The labelling is misleading. It is far easier for the reader to understand this figure if every panel has a heading of either "Europe" or "Arctic". Also every Y axis should have the evaluation metric indicated for every panel ("Bias", "NSE", "r", "Variability"). The caption should also spell out NSE, r (is this Pearson's or Spearman's?) and what is meant by "variability" (is this standard deviation or variance?).

**We have clarified the figure and restructured it along with the other hydrological evaluation figures to follow the same structure.**

Minor text corrections:
p4 line 14 The word "data" is plural. A dataset (singular) contains a lot of data (plural). Hence in both places on this line change "is" > "are".
p4 line 14 "On notable" > "One notable".
p5 line 33 and p6 line 1 "Priestly" > "Priestley".
p8 line 5 "method overestimate" > "method overestimates".
p8 line 5 "The updating method also produce" > "The updating method also produces".
p8 line 7 "a difference already in the observations" > "a difference in the observations".
p8 line 15-16 "is mainly used interim to bridge" > "is mainly used as an interim measure to bridge".
p10 line 6 "for the first about 90-100" > "for about the first 90-100".
p10 line 11 "which is on a much larger magnitude" > "which is of a much greater magnitude" [note the "of" not "on"].

**Thank you for the detailed language checks, which are much appreciated! We will correct accordingly.**

**Response to anonymous reviewer #2:**

**We appreciate very much the reviewer's comments which we will answer below.**

Major comments:
1.) In the introduction the authors claim that "forcing data for large scale hydrological models is essentially not available. . ." – This claim is not correct. In fact there are numerous, (semi) operational global hydrological models that have solved the problem of forcing data in different ways, for example the Global Flood Awareness System, the Global Flood Monitoring System, and the Global Flood Forecasting System. Please refer to their relevant scientific articles on what type of global forcing data they use to derive hydrologic model initial conditions. In addition, see also the recent article from Hirpa et al. (AMetSoc, 2016) on this topic which should be considered by the authors. Finally, also ERA-5 is now available and produced in "near real time". This should also be inclued and discussed.

**Such claims are of course only subjectively correct, and we will rewrite that statement. From our point of view, we need consistent data for temperature and precipitation that are close to observations. This issue has been solved in different ways for different projects with a global approach. Each of the listed systems above have made their own versions of forcing data, but they do not share the data themselves openly as far as we are aware, thus not "available". We will discuss these different global systems and how they have solved the issue in the revised introduction.**
**We will also discuss gridded and satellite products that can be used as forcing, including their pros and cons.**
**Thanks for the Hirpa et al (2016) paper, which we have traced to the paper Hirpa, F. A., Salamon, P., Alfieri, L., Pozo, J. T. D., Zsoter, E., & Pappenberger, F. (2016). The effect of reference climatology on global flood forecasting.** *Journal of Hydrometeorology, 17*(4), 1131-1145. **We will include the results in the revised introduction.**
**Regarding ERA-5, we intend to make use of this once available for a long time period during 2018, and make a new version of GFD based on that. We add this to discussions and outlook.**

2.) Numerous datasets that are claimed to lack temporal coverage have in fact coverage also of recent years including the MSWEP dataset. The manuscript should reflect the latest status of these datasets.

**Our intention with temporal coverage was for the "near real-time", which to us is until at least last month. We will mention the full extent of WFDEI-CRU and MSWEP.**

3.) Furthermore, the authors have omitted completely the TRMM (and now Global Precipitation Measurement) datasets that represent an important source of near-real time precipitation forcing which is clearly the most important variable for the forcing data of hydrological models. Those need to be at least mentioned/referenced with an explanation of why those are not used in this work.

**We have added a section in the introduction where we discuss the different data sets with daily resolution data and a global or near global extend, including CPC-products, TRMM and GPM.**

4.) The authors claim that their method is similar to the method used in the WFDEI dataset. Yet, in Fig. 2d the relative difference between WFDEI and GPCC is considerable whereas the relative difference between GFDCL and GPCC is very small. This suggest that the changes introduced by the authors in comparison to the methodology to WFDEI have a significant impact. Instead of claiming that this is simply due to the use of different precipitation sources this should be further investigated and explained.

**The method is very similar, but the version of GPCC is different between WFDEI and GFDCL, which makes all the difference. We have more clearly distinguished the methodology and the data set itself in the revised version. The methodology is the same as in WFDEI, but the data sets are different, which explains the differences in the end product.**

5.) This manuscript has almost 20 (!) abbreviations. Some of them are spelled out before their first use, others not. Some important ones such as GFDOD1 and GFDOD2 are not properly explained. Even though I am familiar with most of the abbreviations it makes this manuscript very hard to read. The authors should at least include a table with an overview of the most important ones (maybe expand tables 1 and 2) or maybe add them as an annex.

**We have tried various versions of this, and found the current to be most clear in combination with brevity in the text. We will consider other ways to describe the data, and to perhaps remove some abbreviations. The GFDOD1 and GFDOD2 refers to the two separate months of GFDOD, however, as there are only minor differences between these months, we will remove them from the paper completely and only discuss GFDOD.**

6.) What is the effect on hydrological simulations when you update with the new observational data on the 10th of each month? This might lead to a significant discrepancy between simulations done on the 9th and then, with the updated dataset on the 10th. Clearly that represents an issue for hydrological forecasting but is not properly discussed by the authors.

**This is indeed an issue. We present this issue in more detail in the revised version, and describe that the last 3-month history of a simulations changes after the update and that forecasts are not "compatible" in the overlapping window of the forecasts around that date.**

7.) Figure 6 d seems to suggest that there is actually less or at least similar bias in the average upstream runoff difference when compared to 6b and 6c. This seems to contradict Figure 5 where the OD period shows the highest absolute difference. Please explain more in detail why this is the case.

**We have changed the palette so that the bias is more clearly visible, which better emphasizes the differences here.**

8.) The GFD claims to be a global dataset for hydrological models. Yet, the hydrological validation was only performed for catchments in northern latitudes. There is currently no hydrological validation for basins located in tropical climates. The validation should be improved including also basins from these regions.

**We have only made detailed simulations for the two norther regions, however, there are operational forecasts made with HYPE initialised with GFD for more tropical regions, such as Niger. We will link to such operational systems and their performance when published.**

9.) The manuscript lacks a paragraph on future developments.

**A short outlook for 2018 was added, with addition of new data sets as well as moving to ERA-5 instead of ERA-Interim.**

10.) The title of the manuscript should be modified and the authors should define in the text what they mean with "near real time". "Near real time" suggests that data is updated within hours or at least days and not monthly.

**Near real-time is of course in general not well defined and depends on the application. No system can every be "real-time" altough some describe their systems that way. The word "near" therefore works as an alert to the reader that it might not be up to their standards of what "real-time" is. We will therefore keep the title, as it points out the added value of GFD in comparison to WFDEI which is more epidodically updated. To mention but one similar definition; in the precentation of GLOFAS in Alfieri et al. (2013), ERA-Interim is introduced as near real-time data, with its three month delay.**

**Below, we silently accept all changes unless commented.**

Minor comments:

- Please add the datasets used for WFDEI to Table 2

- Please add a reference for GHCN-CAMS into the references

**Its already there: Fan and van den Dool (2008)**

- P.10, line 14: last sentence is unclear. Please describe further and rephrase

- What is the difference between GFDOD1 and GFDOD2?

**We remove these as described above.**

- The nonlinear scale in Fig. 2 and 4 makes it very difficult to look at the results. Basically everything greater than +-25% is hardly distinguishable. Please choose a different scale or use the one applied in Fig. 6 (and possible also a different color scale)

**We have revised all the figures for better color palettes, now using colorbrewer suggestions, and moved to a linear scale for most plots.**

- In the section on "Meteorological evaluation" the authors write ". . .and focus instead on comparisons to the WFDEI dataset." However, in the following evaluation you compare the GFDCL mostly with GPCC, EI or OD. Please clarify.

- Does Fig. 2 show the relative difference of EI, GFDCL and WFDEI to GPCC7/CRUts? Please make this more clear.

- Is the precip bias between EI and GPCC7 in line with other studies looking at the precip bias of EI? If yes please add the relevant reference.

**We do not see what such a reference would add. Our method is clearly defined and there is no room for other studies showing different results.**

**Near real-time adjusted reanalysis forcing data for hydrology**

Peter Berg[1], Chantal Donnelly[1], and David Gustafsson[1]

[revised manuscript text omitted]

# 3   Results

We begin with evaluating the GFDCL data set, as well as comparing differences between the various [c3]HydroGFD versions.
Thereafter, we present analysis of hydrological simulations for Europe and the Arctic.
* * *
[c7] *Text added.*
[c8] *Text added.*
[c1] *Text added.*
[c2] *Text added.*
[c3] *Text added.*

**3.1 Meteorological evaluation**

**Climatology 1979–2013:** GFDCL is directly comparable to the WFDEI data set due to the very similar method[c4], but will differ due to different underlying data, and handling of precipitation under-catch. Because WFDEI was on several occasions evaluated against flux tower measurements across the globe (Weedon et al., 2011, 2014; Beck et al., 2016), we do not repeat such evaluation for GFDCL here, and [c5]compare instead [c6] to the WFDEI [c7]and other data set[c8]s.

[revised manuscript text omitted]